# Process evaluation and exploration of telehealth in motor neuron disease in a UK specialist centre

Esther Hobson ,[1,2] Wendy Baird ,[3] Mike Bradburn ,[4] Cindy Cooper ,[4] Susan Mawson ,[3] Ann Quinn,[5] Pamela J Shaw ,[1,2] Theresa Walsh,[1,2] Christopher J McDermott [1,2]

► http://dx.doi.org/10.1136/bmjopen-2018-028525

For numbered affiliations see end of article.

**Correspondence to**
Dr Esther Hobson;
e.hobson@sheffield.ac.uk

## ABSTRACT

**Objectives** To evaluate the processes involved in using a novel digitally enabled healthcare system (telehealth in motor neuron disease (TiM)) in people living with motor neuron disease (MND) and their informal carers. We examined TiM implementation, potential mechanisms of impact and contextual factors that might influence TiM implementation or impact.

**Design** An 18-month, single-centre process evaluation within a randomised, pilot and feasibility study.

**Intervention** TiM plus usual care versus usual care alone.

**Setting** A specialist UK MND care centre.

**Participants** 40 patients with MND and 37 primary informal carers.

**Primary and secondary outcome measures** Patient, carer and staff outcomes and experiences using semistructured interviews. Descriptive data on implementation and use of TiM.

**Results** The TiM was acceptable and accessible to patients, carers and staff. Intervention uptake and adherence were good: 14 (70%) patients completed a TiM session at least fortnightly. Barriers to TiM use (such as technology experience and disability) were overcome with well-designed technology and face-to-face training. Reported potential benefits of TiM included improved communication and care coordination, reassurance, identification of complications and the potential for TiM to be an alternative or addition to clinic. Benefits depended on patients' current level of needs or disability. The main challenges were the large number of alerts that were generated by TiM, how the clinicians responded to these alerts and the mismatch between patient/carer expectations and nurses actions. This could be improved by better communication systems and adjusting the alerts algorithm.

**Conclusion** TiM has the potential to facilitate access to specialist care, but further iterative developments to the intervention and process evaluations of the TiM in different services are required.

**Trial identifier number** ISRCTN26675465.

## BACKGROUND

Motor neuron disease (MND) causes progressive muscle paralysis, severe disability and eventually death within on average 2 to 4 years.[1] Attendance at a specialist multidisciplinary

<div style="border:1px solid #000;">

### Strengths and limitations of this study

► This is a detailed evaluation of a digitally enabled care system that aims to provide accessible, specialist holistic care to patients with motor neuron disease and their carers.
► The use of mixed methods provided an in-depth understanding of the processes involved in the new care service, the potential value of telehealth in motor neuron disease and factors that are likely to influence its use, implementation and success.
► This was a small, single-centre study using only one nurse to deliver the intervention.

</div>

clinic (MDC) is associated with improved survival and use of assistive aids, devices and proven therapies.[2–7] Patients face many barriers accessing specialist services, meaning many patients fail to gain the full benefit from these services.[8] There is some evidence that digital-enabled technologies can improve access to specialist care.[9] Video-consultation (telemedicine) is used in some centres and appears to be feasible and acceptable.[10–14] However, between consultations, changes in patients' condition may not be recognised. There are several technologies in development that can frequently record aspects of MND progress (telehealth) such as speech or hand function. They have predominately been developed for research, but telehealth could be used for clinical care purposes.[15–18] No digital system has yet been evaluated that can monitor the full range of problems faced by patients with MND in order to detect and act on issues that arise in a timely manner.

There also remains a need to develop interventions to support carers of those with MND. Medical interventions such as non-invasive ventilation and gastrostomy feeding require carers to conduct complex tasks with little training.[19] Carers report that these interventions are associated with increased physical

strain and carer time.[20–22] Provision of disease-specific information and signposting to external support services have been identified as helpful[23 24] and while one small study of intensive case management reported positive feedback,[25] another showed no improvement in carer strain.[26]

We employed a user-centred design process to develop the telehealth in MND (TiM) telehealth system.[27] This allows patients and carers to send information about their condition to the specialist care team.[27] Early testing has already been conducted with a small number of potential users, but further testing in the clinical environment was required.[27] Telehealth services are complex interventions, consisting of different component parts and whose success relies on the context and individual behaviours of those using and delivering the intervention.[28] Clinical trials of complex interventions such as telehealth face problems with patient and staff engagement and often lack of understanding of how the intervention is used, what works and for whom.[29–31] People living with MND and their carers face additional barriers to using digitally enabled care due to their disability and age. The complexity of the disease and the holistic aims of care in this terminal disease mean telehealth systems will need to differ from more simplistic, target-driven care systems (such as blood pressure management). With these challenges in mind, the Medical Research Council recommend that development of a telehealth service is iterative with learning from development, piloting, evaluation and implementation informing each step.[32]

## Aim of the study

We conducted a pilot and feasibility randomised controlled trial (RCT) in patients and carers of usual care plus TiM versus usual care. The results of the pilot and feasibility trial are described in a parallel publication.[33] In this paper, we describe the process evaluation,[34] which aimed to gain a deeper understanding of the use of the TiM system.

We aimed to observe the processes that occurred when TiM was used to deliver specialist multidisciplinary care in MND exploring:

► TiM technology set-up and delivery.
► Participants' adherence to TiM.
► Participants' attitudes towards TiM.
► Clinicians' attitudes towards TiM.
► Potential impacts of the TiM on participants and staff (both intended and unintended).
► The mechanisms and contextual factors that may affect the impact and implementation on a larger scale.

We also aimed to:

► Identify problems with TiM.
► Begin to implement and evaluate improvements.
► Identify further improvements and uses of the TiM.

## METHODS

### Study design

The methods of the pilot and feasibility RCT are described in a parallel publication.[33] A supplementary file (online supplementary file) contains the TiM protocol, topic guides and statistics analysis plan. In brief, patients with MND (amyotrophic lateral sclerosis, progressive muscular atrophy and primary lateral sclerosis) and their primary informal carer currently receiving care from the Sheffield MDC were recruited and randomised to receive either the intervention (TiM plus usual care) or the control (usual care alone). Participants were followed up for between 6 and 18 months. Usual care involved invitations to the MDC 2 to 6 monthly plus access between visits to the multidisciplinary team (MDT) via an MND specialist nurse by telephone, email or through liaison with other healthcare professionals.

### Ethics and consent

Written or witnessed verbal consent was gained from patients, carers and the telehealth nurse prior to randomisation.

### Intervention

A detailed description of TiM has been published.[27] The TiM patient/carer app contains a set of patient and carer questions assessing the disease progression, complications of MND, use of medical interventions and patient and carer well-being. The app was loaded onto a 7-inch Samsung Galaxy tablet with 3G internet. We asked participants to complete the questions weekly or more frequently if they desired. The results were sent using the internet to a clinical portal used by the MDC specialist nurse (the telehealth nurse) who had over 15 years experience of MND care. The portal assigned a red, amber or green flag to all the answers depending on a predetermined algorithm created by the TiM developers. The telehealth nurse was expected to review the answers and could liaise with specialists in the wider MDT and provide telephone advice. All patients and carers continued to receive usual care, Sheffield MDC appointments and telephone/email access to the telehealth nurse. Clinic appointments could not be postponed but could be expedited. The nurse was required to have a discussion with the patient, carer or clinician to confirm the accuracy of the information prior to making any clinical decisions. Also loaded onto the tablet were educational materials (known as the Knowledge Centre) and within the app was a section where patients could make a list of issues the participant wished to discuss at clinic (the Problem List) and educational messages, which appeared between questions.

EH recruited participants at home. Participants were given the tablet and Wi-Fi-enabled standing weighing scales. They were given face-to-face training, written information, telephone and email support. EH contacted patients after 2 weeks if they did not submit a TiM session and visited at 1 and 6 months for interviews, which provided an opportunity to resolve technical issues. EH

trained clinicians to use the clinical portal in face-to-face training, which took approximately 10 minutes. During the trial, we gained feedback from users in order to improve the system. The trial started in October 2014, and one major change to the patient app and the clinical portal was implemented in October 2015.

### Data collection

A detailed description of all data collection is described in the accompanying feasibility paper[33] and in the protocol (online supplementary file). Patient-/carer-reported outcome measures were collected at 0, 3, 6, 12 and 18 months using postal questionnaires. These assessed patient disability, depression, anxiety, quality of life and health utility. To gain a deeper understanding of the processes involved, we evaluated patient, carer and staff experiences of the TiM. Patients and carers completed satisfaction questionnaires using postal questionnaires at the same intervals. We also collected clinician-reported adverse events and health resource use using postal questionnaires.

We conducted semistructured interviews. Interviews were conducted by EH at baseline (control participants) and 1 and 6 months (intervention participants). Interviews with the telehealth nurse were conducted at 14 and 18 months. The 18-month interview was conducted by an experienced qualitative researcher WB. Additionally, an interview with an MND specialist community nurse who cared for a number of participants in both treatment arms during the study was conducted at 18 months. Topic guides (online supplementary filepage 41) included participants' experience of specialist MDC and community care, attitudes towards technology and digitally enabled care, expectations and experiences using TiM, reasons for good and poor TiM usage, potential impacts TiM could have on MND care, and improvements and future applications of TiM. Early results and observations from the trial influenced the topic guides for later interviews. Interviews were audio-recorded, transcribed verbatim, checked by the interviewer and organised using NVivo.[35] Thematic analysis was used.[36] A triangulation process compared the quantitative and qualitative data to further understand and explain important, incongruent and unexpected observed phenomenon.[37] EH conducted the analysis with supervision from WB. Results were presented to the trial management group and steering committee.

MDC physicians completed a satisfaction questionnaire ('Shadow monitoring form', see online supplementary file page 71) when they saw a patient in clinic using TiM capturing opinions on the accuracy, feasibility and acceptability of TiM. Field notes were also taken to describe any problems with TiM, solutions that were used and their outcomes. At the end of the trial, we downloaded all data collected by TiM into Excel. This contained the time and date of every session, all answers provided by patients and carers, all alerts generated by the system and all clinical notes made by the telehealth nurse. We used descriptive statistics to report the TiM use. We attempted to record the telehealth nurse's time spent using TiM using diaries but these were incomplete. Instead, this was explored using interviews.

### Patient and public involvement

During development of TiM[27] and the protocol, we consulted patients, carers and the Sheffield MND Research Advisory group (a patient and public involvement group). They reviewed the intervention, principles of the trial, trial design, outcome measures and participant information leaflets and provided comments on their feasibility and accessibility. They were not involved in recruitment. Results of the study have been communicated at various public meetings, through the Sheffield MND Research Advisory group and local branch of the MND Association and a lay summary will be circulated. Members of this group attended the trial steering and trial management groups. AQ was a member of the trial management group, provided advice on the research methods, interpretation of the data and dissemination and is a co-author on this paper.

## RESULTS
### Participants

Forty patients and 37 informal carers (three patients had no carer) were recruited. Two TiM patients withdrew due to severe illness and one died before 6 months, the rest used TiM for between 6 and 18 months. Participant characteristics are reported in full in the parallel paper.[33] The patients broadly represented those attending an MDC including patients at all stages of disease (table 1). While patients' use of digital technologies was common, some participants had little or no experience using technology. Thirty-eight (95%) patients had home broadband and 33 (83%) had mobile internet reception.

### TiM technology set-up and delivery
(Online supplementary data file tables 1–4)

All participants allocated the intervention received the TiM system and could use it using mobile or home broadband. Technical problems and solutions were all resolved and had little impact on the trial except the Wi-Fi-enabled scales, which were not found to be practical due to a lack of reliability connecting to the internet (table 2). At 6 months, all the patients and 14 (93%) of the carers felt TiM was easy to use (figure 1). Fourteen (93%) patients and carers did not think the system was tiring and 13 (87%) did not think the questions were distressing, nor the use of the system intrusive.

'It's so easy to do; it literally takes five minutes from home.' Patient 317

Twenty-seven patients (68%) reported some difficulty using technology due to upper limb disability but 34 (85%) could use the patient app independently (although they might need a carer to help charge the tablet). One

**Table 1** Participant characteristics

| | Telehealth (n=20) | Control (n=20) |
|---|---|---|
| Patient gender, male | 14 (70%) | 14 (70%) |
| Age (years) Mean (SD), range | 60.4 (11.7), 30–78 | 60.0 (10.0), 39–73 |
| Disease duration (months) Mean (SD), range | 53 (48), 12–197 | 46 (35), 7–123 |
| King's ALS clinical stage* | | |
| 1 | 3 (15%) | 2 (10%) |
| 2 | 4 (20%) | 5 (25%) |
| 3 | 5 (25%) | 8 (40%) |
| 4 | 8 (40%) | 5 (25%) |
| Use of the TiM app | | |
| Independently | 17 (85%) | 17 (85%) |
| Help from carer | 1 (5%) | 1 (5%) |
| Patient instructs carer | 2 (10%) | 2 (15%) |
| Patient technology use† | | |
| Daily | 14 (70%) | 18 (90%) |
| Few times per week | 3 (15%) | 1 (5%) |
| Once a week | 1 (5%) | 1 (5%) |
| Every few weeks | 0 (0%) | 0 (0%) |
| Never | 2 (10%) | 0 (0%) |
| | Telehealth (n=19) | Control (n=18) |
| Carer gender, male | 4 (21%) | 5 (28%) |
| Carer age (years) Mean (SD), range | 59 (12), 42–84 | 60.8 (11), 38–73 |
| Relationship to patient | | |
| Partner | 18 (95%) | 16 (89%) |
| Child | 0 (0%) | 1 (6%) |
| Parent | 1 (5%) | 1 (6%) |
| Carer technology use† | | |
| Daily | 12 (67%) | 16 (84%) |
| Few times per week | 1 (6%) | 0 (0%) |
| Once a week | 1 (6%) | 2 (11%) |
| Every few weeks | 0 (0%) | 1 (5%) |
| Never | 4 (22%) | 0 (0%) |

*King's stage 1 refers to patients with functional deficit in one domain, stage 2 refers to disability in two domains, stage 3 refers to disability in three domains and stage 4 refers to patients requiring NIV and/or gastrostomy. King's stage was calculated using the ALS-FRS-R scale at baseline.
†Technology use: computer, tablet, smart phone.
ALS-FRS-R, Amytrophic lateral sclerosis functional rating scale revised; NIV, Non-invasive ventilation; TiM, telehealth in motor neuron disease.

patient could use the app but during training it was noticed that his answers were not accurate (Patient 073). He was unable to communicate verbally and exhibited behaviours suggestive of mild cognitive deficits. In this case, he was happy for carers to help him use it.

Participants found face-to-face training important. Many were pleasantly surprised by their achievements.

'I thought I wouldn't be able to do it (laughter) but I can….' Carer 228

The main perceived barrier was participants' experience using technology.

'…we're quite happy to deal with it, as long as it wasn't too techy' Carer 062

Many participants (including those using technology daily) perceived themselves lacking an intrinsic ability to use technology, saying that they were 'bad' at it, thinking others were more 'wired' to using technology. They shared common attitudes towards their abilities, experience and approach to technology. They reported struggling to learn to use new technology finding it stressful and were fearful of making a mistake or breaking the device. They did not feel in control of technology, expressing frustration when technology did not 'obey' them. This meant they were not confident adapting to unfamiliar technology. They tended to use technology for a limited number of basic tasks, lacking confidence to problem solve when technology 'went wrong', relying instead on others. Some low users explained they never learnt, or did not see the need to use technology, particularly if their partner used it.

One couple (Patient 166) had no experience of technology and were highly critical of the perceived intrusion of technology in their lives. They described themselves as 'technophobes'. However, they trusted their care team would provide them with a secure and safe system. Only one carer in his 80s (Carer 217) struggled to 'catch on', needing help from his partner. He had little prior experience with technology, and it was later noted that he had difficulties with language due to another medical condition.

## Participant adherence to TiM

(Online supplementary data file tables 4–6)

Participants preferred to complete the TiM weekly (13, 87% of patients; 8, 53% of carers). They reported that weekly sessions helped them remember to use the TiM. Some suggested they would complete TiM less frequently if nothing had changed. The telehealth nurse suggested that fortnightly information was sufficient to enable her to assess a patient's progress and even less frequently for carers. Adherence was calculated by dividing the total number of sessions completed by the number of weeks participants were in the trial (figures 2 and 3). Fourteen (70%) patients and 10 (55%) carers completed TiM sessions on average, at least fortnightly. Thirteen (70%) carers completed sessions, on average, at least every 3 weeks. Some participants who completed fewer sessions still continued to use TiM regularly. Adherence dropped during the study, but at 12 months 80% of patients were still using it weekly. The main reasons for low adherence were forgetting, holidays and deterioration in their illness. A lack of feedback about the answers participants

**Table 2** Technology problems encountered during the trial

| Problem | Solution adopted | Impact on the use of the TiM | Recommendations for future TiM use |
|---|---|---|---|
| **TiM Patient App software** | | | |
| Poor finger dexterity | Handheld stylus provided | 85% used TiM independently. | Provide stylus to all patients. |
| | Carer help to use device | Interview data suggested that help from carers was acceptable. | Encourage carer support. |
| Difficulties entering login password | Telephone support | Problem resolved with second TiM app release. | Provide face-to-face training. Provide telephone technical support. |
| | Login page redesigned | No further problems reported. | Make local staff familiar with the software. |
| Lack of confidence using the app or other features on the TiM | Face-to-face training Partner/family helped | None: all participants could use the app. | Provide face-to-face training plus an additional contact after a few weeks to reinforce learning. Identify low confidence/experience users and provide extra training. |
| Patients not giving correct answers | Patient completing TiM with their family | Uncertain impact. | Capacity assessment at recruitment. Check TiM answers in training and in clinic. |
| **Tablet** | | | |
| Tablet stored in place not accessible to patient | No solution available | Interviews suggested that adherence was reduced for two patients. | Use patients' own equipment where possible. |
| Tablet battery drained, unable to switch on | Telephone advice | No impact. | Use patients' own equipment. Familiarise local staff with hardware. |
| Unexpected screens/software updates | Telephone advice | Reduced user confidence in tablet but no impact on use. | Use patients' own equipment. Use a basic tablet that only displays the TiM app. |
| User fear of 'breaking' the tablet | Face-to-face training to improve user confidence | Patients/carers reluctant to use the additional features on the tablet. | Use patients' own equipment. Face-to-face training. |
| **Internet connection** | | | |
| Poor 3G phone signal | Used patients' own broadband | None: solution found for all patients (3G or patients' own broadband) | Use patients' own broadband. Check internet availability prior to TiM enrolment. |
| Patients switched Wi-Fi off | Home visit required | Several TiM sessions failed to download. | Monitor adherence. Alert team if adherence is low. |
| Unreliable connection between scales and tablet/broadband | A separate 3G Wi-Fi route provided ('Mifi'). No solution if using broadband | Additional home visits required and loss of weight data for several weeks. | Manual weight recording. Avoid using peripheral devices. |
| **Clinician portal** | | | |
| Password/login problems | Support by external IT team | Delayed access during MDC visit. | Local systems access support. |

MDC, multidisciplinary clinic; TiM, telehealth in motor neuron disease.

gave was also a disincentive to use TiM, particularly when they were expecting feedback.

'It's like all surveys… they say "your opinion is important…" You say something absolutely and nobody comes back to you. And you think: well how important is that survey?' Carer 228

The use of the additional features of the tablet (the educational messages, knowledge centre or problem list) was not monitored during the trial. Unlike the main TiM app, participants reported that they were less likely to use these. Most were aware of these resources, particularly the short messages that appear while completing the TiM questions. Some participants reported reading some of the additional material, but no one felt strongly that these had had a major positive impact with only half agreeing that they were useful (figure 1).

## Participants' attitudes towards TiM
(Online supplementary data file tables 7–11)

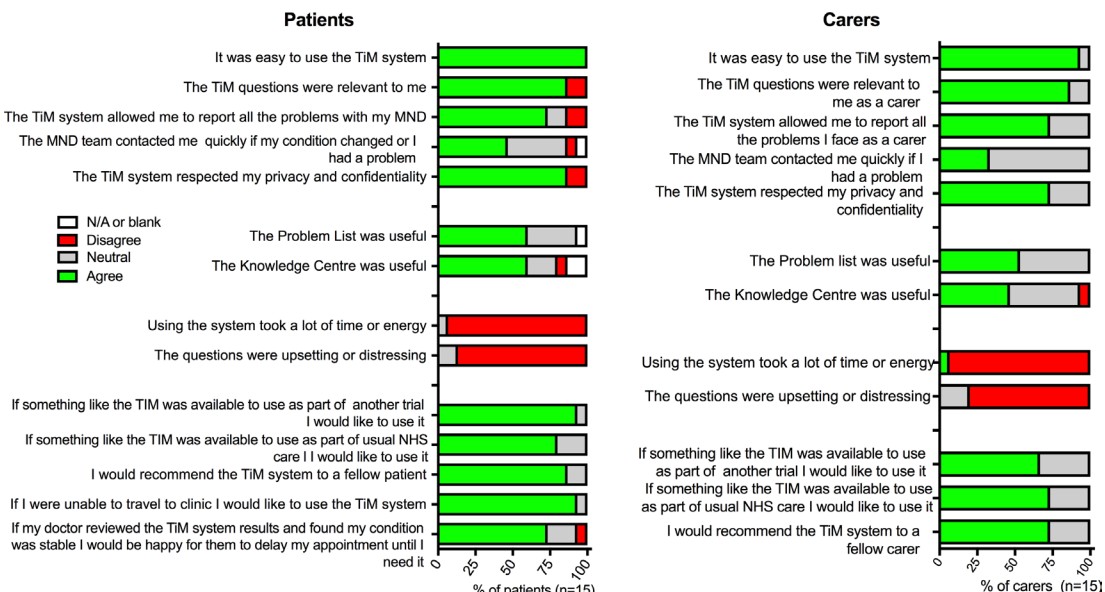

**Figure 1** Patient and carer satisfaction with TiM telehealth at 6 months. MND, motor neuron disease; TiM, telehealth in motor neuron disease.

Fourteen (87%) patients and carers agreed that TiM questions were relevant to them and 12 (73%) patients and carers felt able to report all the problems they experienced using TiM (figure 1). Thirteen (80%) patients and 12 (73%) carers would use it again. Fourteen (87%) patients and 12 (73%) carers would recommend it to another patient.

'I think it's probably one of the best ideas to come out of the NHS for years.' Patient 122

Participants did not find using TiM intrusive or upsetting. They explained they were reminded about their

disease everyday already and knew themselves if they had deteriorated without using the TiM.

'It doesn't bother me. When I was first diagnosed I was not keen on talking about it. I've since got better and I think that helps. I'm quite happy to be reminded. I think you are reminded every day. That tablet makes no difference to that.' Patient 381

Some participants thought the questions were repetitive, particularly carers. Some noticed they reported the same problems every week. Some felt the questions were insufficiently sensitive to reflect the day-to-day fluctuation

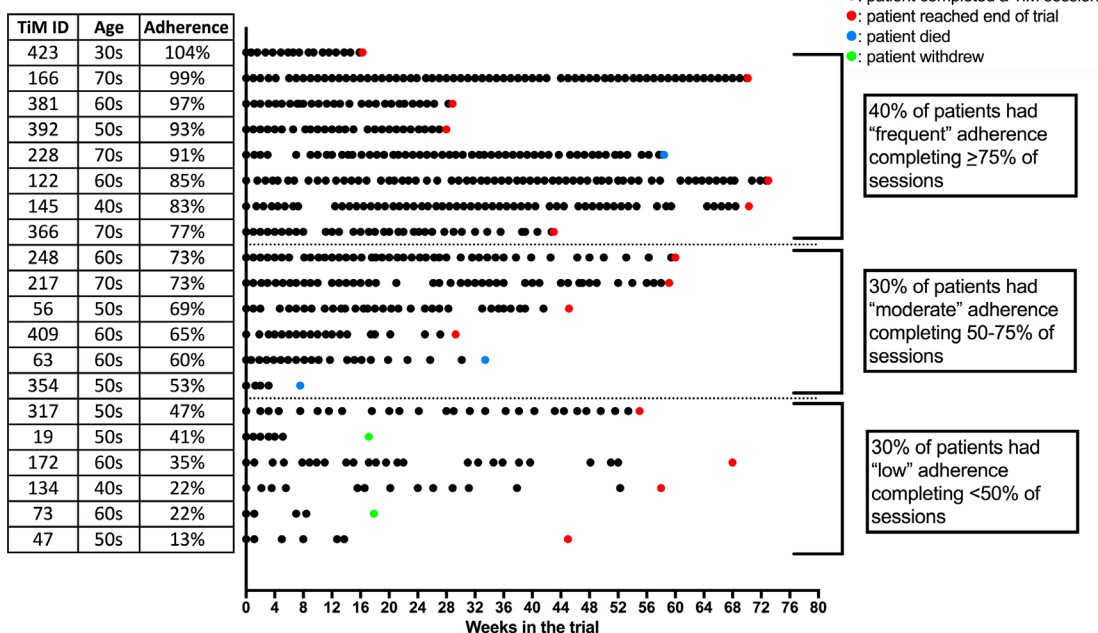

**Figure 2** A visual representation of individual patient adherence over the duration of the trial. Each black dot indicates a completed TiM session. A coloured dot indicates the end of follow-up. TiM, telehealth in motor neuron disease.

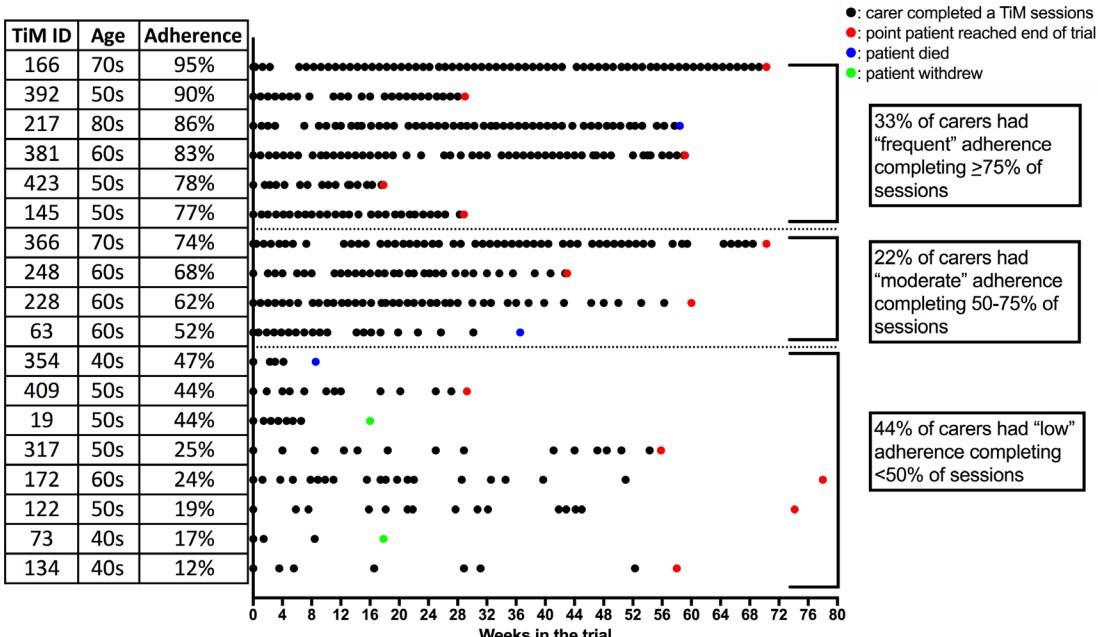

**Figure 3** A visual representation of individual carer adherence over the duration of the trial. Each black dot indicates a completed TiM session. A coloured dot indicates the end of follow-up. TiM, telehealth in motor neuron disease.

in their condition. One carer suggested that the questions should use other ways to assess carers lives more critically, which could be used to encourage carers to consider their own well-being and adopt coping strategies.

### Clinicians experiences using TiM

(Online supplementary data file tables 12–13)

The telehealth nurse said that using the clinical portal was 'very, very easy', taking 'minutes' per use. She explained that initially she looked at the system every day but by the end of the trial she looked weekly and sometimes a little less often. She would look at each patient but focused on new and changing alerts. She explained that if she saw a patient/carer's answer was associated with a green flag it indicated everything was 'ok', orange meant 'there's maybe some elements that might need to be looked at', while a red flag indicated a problem.

#### Safety and accuracy of TiM

No adverse events related to TiM were reported, and no participant reported delaying seeking medical attention as a result of TiM. Thirty-eight clinician satisfaction forms were completed when patients using TiM attended clinic. All agreed that TiM gave a useful and accurate picture of the patient's and carer's condition and was a positive influence on the consultation. However, 20 (54%) were completed by the investigator (EH) who saw the patient in clinic, 18 (46%) were completed by other physicians.

### The reported potential impacts of TiM

(Online supplementary data file tables 14, 15)

Reported potential impacts included improved monitoring and communication with the clinical team and improved effectiveness of MDC visits and care coordination. Other potential impacts included improved patient

awareness of their disease and the ability to identify carer distress.

### Improving links with the clinical team

Participants thought TiM could provide a 'direct link' with the care team, increasing monitoring, providing better connection with specialists and enabling earlier identification of problems. Participants (particularly carers) also felt reassured that they were being thoroughly monitored. Reassurance was particularly beneficial for those who had infrequent contact with clinicians, either because they were early in the disease or were progressing slowly.

> 'Patient: … for the first year no day was normal… You imagine symptoms, … you think…that must be related to the MND…?
>
> Interviewer: Do you think there might have been a point in your disease where those questions …were useful?
>
> Patient: Nearer the beginning, definitely…If I could have camped in (neurologist)'s house for the first six months I would have done, just so she was there, so I could say; "but what about this, what about that?"… in the first year I would have filled that in every day, just to have that touch point' Patient 047

### Improved awareness of the disease

Many participants thought that TiM improved their awareness of their condition. They felt that they did not need to track their progress using TiM as they knew themselves how they were changing, but some had become more aware of their weight and nutrition. Rapid progression was something all patients feared, so those with

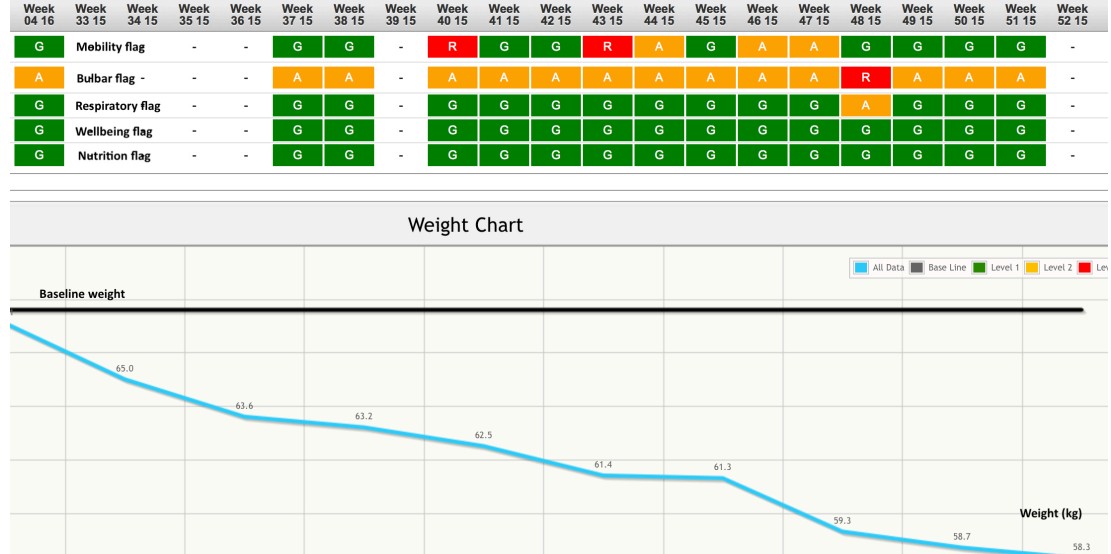

| | Week 04 16 | Week 33 15 | Week 34 15 | Week 35 15 | Week 36 15 | Week 37 15 | Week 38 15 | Week 39 15 | Week 40 15 | Week 41 15 | Week 42 15 | Week 43 15 | Week 44 15 | Week 45 15 | Week 46 15 | Week 47 15 | Week 48 15 | Week 49 15 | Week 50 15 | Week 51 15 | Week 52 15 |
|---|---|---|---|---|---|---|---|---|---|---|---|---|---|---|---|---|---|---|---|---|---|
| G | Mobility flag | - | - | G | G | - | R | G | G | R | A | G | A | A | G | G | G | G | - |
| A | Bulbar flag - | - | - | A | A | - | A | A | A | A | A | A | A | A | R | A | A | A | - |
| G | Respiratory flag | - | - | G | G | - | G | G | G | G | G | G | G | G | A | G | G | G | - |
| G | Wellbeing flag | - | - | G | G | - | G | G | G | G | G | G | G | G | G | G | G | G | - |
| G | Nutrition flag | - | - | G | G | - | G | G | G | G | G | G | G | G | G | G | G | G | - |

**Weight Chart**

Baseline weight

All Data ■ Base Line ■ Level 1 ■ Level 2 ■ Lev

65.0
63.6
63.2
62.5
61.4
61.3
59.3
58.7
Weight (kg)
58.3

**Figure 4** The 'heat map' for Patient 409. This is a screenshot from the clinical portal. The heat map reports the highest level flag for each section of the questionnaire (mobility, bulbar, breathing, well-being, nutrition) at each week. This indicates the patient has red and amber flags for both bulbar and later mobility. Below is the weight compared with the patients' weight at baseline demonstrating a greater than 15% weight loss.

slowly progressive disease noticed that their answers did not change rapidly. This made them feel more positive.

### Early identification of problems

The telehealth nurse described TiM as 'a tool to aid the patient and then aid the nurse'. She could use TiM to gain more information and alert her earlier to problems. In one example, the TiM detected that one patient's condition was declining quickly and the carer reported high levels of strain. The couple was struggling to accept his decline, and the patient tried to minimise his symptoms. TiM alerted the telehealth nurse to this patient's rapidly declining weight and bulbar function (figure 4). She contacted him, discussed this information and encouraged him to consider gastrostomy insertion, which he accepted. He was asked about whether TiM affected this decision and explained that the information had been helpful:

> 'The questions nudged me to facing what I could do and not what I can't…I was frightened by the speed of loss of weight but was convinced how much muscle I lost. [sic]' Written quote Patient 409

Other patients had reported problems on the app that they had not disclosed to their care team. The telehealth and community nurse speculated that this was because the patients were struggling to accept the change in their disease or the need for treatment and might have felt that they can be more honest using technology. They found this information useful to gain a more complete picture of the situation.

### Improving the MDC

The telehealth nurse and physicians suggested that TiM could improve the MDC. The telehealth nurse said if

TiM could reduce the burden faced by patients attending clinic, it would be welcomed by nurses. Potential benefits included facilitating sharing of information, collecting information and identifying problems before the visit. This could help plan the MDC visit to address the pertinent issues, shorten the time spent in hospital and make telephone appointments easier and a better alternative to face-to-face.

After seeing patients in their clinic using TiM, clinicians completed a feedback form (Shadow monitoring protocol). Twenty-three forms were completed. On 19 (79%) occasions, clinicians agreed that the information gathered by TiM could be used to make appropriate decisions without the patient attending in person. After 4 (17%) appointments, they answered 'neutral'. Fourteen (93%) patients said they would use TiM if they were unable to travel to clinic and 11 (73%) felt that they would be happy to have their appointment delayed if the doctor felt they were stable. Three (20%) were unsure and one (7%) disagreed. When interviewed, participants thought that the individual should choose how they used TiM depending on their needs, preferences and speed of progression. Some patients who were progressing slowly did not feel frequent MDCs were valuable. They were happy to use TiM and reduce the frequency of appointments as long as they could access the team quickly if required. Others wanted to attend the MDC to receive psychological support or address their problems and did not want to reduce visits. Some were not keen on telephone appointments, particularly when discussing sensitive matters or when they had speech difficulties. Similarly, the telehealth nurse explained face-to-face meetings helped her develop her relationship and gain information about the patient and carer. She felt some

aspects of their care would need a combination of TiM system and communication with the patients/carers and other members of the MDT. Both clinicians and patients identified the need for face-to-face consultation to fully evaluate respiratory symptoms.

### Impact on carers

Carers felt that their well-being was directly linked to the patients' well-being: interventions that improved the patient's life or carer's ability to provide care could improve a carer's well-being. Carers felt reassured that their loved one was being monitored and that problems could be identified and solved more rapidly.

'I think the benefit to (patient) is real. Because … somebody is there on hand looking at things… Because it's slow with P and he doesn't need as much attention and care, it's easy to feel detached from any positive interaction. Whereas with (TiM), you know somebody's there and if there was something you'd pick up quite quickly as apposed to, waiting until your next twelve week appointment' Carer 122

Carers and the telehealth nurse reported that the information highlighted by TiM system could help couples accept the diagnosis and the problems they were facing and enable them to look positively towards receiving additional medical and social care. Carers reported that TiM prompted them to consider their own well-being and consider help if they were experiencing difficulties. They explained that usually the patient was present in meetings and healthcare professionals tended to focus on the patient's needs. This meant it was difficult for them to discuss their own concerns because of the potential impact this may have on the patient. Carers thought that TiM provided an opportunity for them to express their feelings in private, separately from the patient but still in the convenience of the home environment, without having to leave the patient. The impersonal nature of TiM was also an advantage as it enabled carers to be more honest about their difficulties.

'I think it's a really good way of doing it. Because whereas I probably try and flower things up a lot of the time …I found that, because it was just me and the tablet and I was able just to be totally honest about how I was… feeling at that particular time… The impersonal format… of the way it was actually done… for me, has been a real help just to be able… to do that.' Carer 409

### Problems with TiM

(Online supplementary data file tables 16–18)

While the participants did react positively to TiM, many felt it had not had a significant impact on their care. At 6 months, only seven (47%) patients and five (33%) of carers agreed, 'The MND team contacted me quickly if my condition changed or I had a problem'. In some cases, participants felt their condition had not

changed during the trial so TiM had not highlighted any problems. In other cases, problems were reported and help was offered but declined either because the participant did not want it or because they did not think the MDT could help. However, the main explanation for this perceived lack of impact related to the large number of alerts generated by TiM, the way in which the telehealth nurse acted on information and lack of interaction between the participants and the telehealth nurse when problems occurred.

### TiM system alerts

Of the total 585 TiM sessions completed, only 19 (3%) of all the patient/carer sessions return answers all of which were flagged as green. Three hundred and twenty-two (55%) sessions contained answers that generated at least one red flag and 244 (42%) generated amber alerts (figure 5A). We analysed individual answers and clinical notes for 10 randomly selected patients (334 sessions). Bulbar questions caused most red alerts (usually due to swallowing difficulties or excessive saliva), followed by limb function (usually due to falls) and respiratory symptoms. In the 334 sessions analysed, we identified 99 notes made by clinicians. Thirty-two (32%) notes reported the telehealth nurse taking action (figure 5B): commonly providing telephone advice or liaising with the MDT to share information and coordinate care. Seventeen (17%) times she planned to review the problem when the patient next came to clinic. In over half of occasions, the telehealth nurse documented that she took no action and in only 18 (18%) occasions did she contact the patient or carer. She explained that she usually took no action because she already was aware of the problem, the participant was awaiting treatment or had declined to accept the recommended treatment/advice. She felt that she was seeing the same problems repeatedly. This was particularly frustrating when she had already taken action or when she felt there was no action required. While she did not feel TiM took up a lot of her time, these multiple problems for which she saw no solution caused an element of psychological burden. She could not control how the alerts were generated and would prefer to be able to 'pause' these alerts.

### Patient-nurse interaction and feedback

Some participants recalled the telehealth nurse contacting them after reporting a problem on the TiM. The nurse would usually offer reassurance or new advice. While the telehealth nurse felt that these interactions did not have an impact, in the face of a relentlessly progressing disease, these interactions were welcomed by patients and carers and it reinforced their perception that specialists were monitoring them closely.

Patient…I did the second questionnaire, and within a day [Telehealth nurse] called.

Interviewer: Were you expecting her to call?

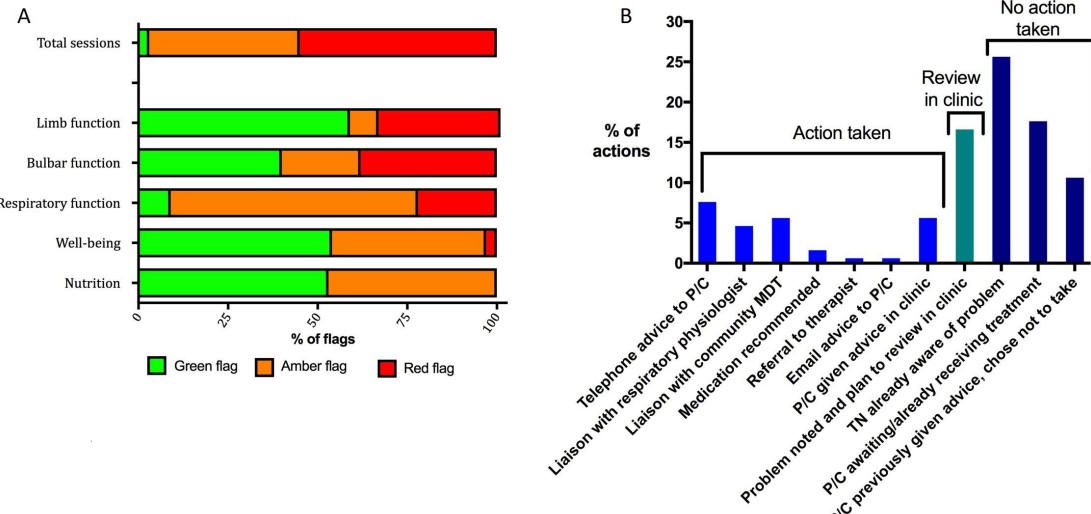

**Figure 5** (A) (left) The frequency of flags generated by the TiM sessions. The total sessions report the frequency of 'top level' flag (green, amber, red) for all 585 sessions completed. Below are the frequencies of 'section flags' generated by 10 patients who completed a total of 334 sessions. (B) (right) Actions described in 99 clinical notes. C, carer; MDT, multidisciplinary team; P, patient; TiM, telehealth in motor neuron disease; TN, telehealth nurse.

Patient: No, I wasn't actually. It was just a bolt out of the blue…I find that quite positive. It shows that the whole idea of it works.' Patient 122

The telehealth nurse prioritised problems that were unexpected and placed less priority on problems she expected to occur in MND. Contacting the patient depended on whether the problem needed or had a solution.

'She red flagged that she'd fallen, which is quite a common occurrence on a lot of patients, and I don't particularly worry unless they've been very, very well and then suddenly.' Telehealth nurse

Problems the nurse perceived not to be significant were often felt to be important to patients and carers. For them, acknowledgement was helpful, even if no solution was needed.

'Patient: (falls) … knocks your confidence …I probably were putting too much onus on (the hospital MDC) because (laughs) we've got this and there's not (much) they can really do about this and we know that…

Interviewer: But that kind of acknowledgement's quite important?

Carer: I do, yeah, it's a bit of support, in't it, it's knowing that somebody else is in your corner.' Carer 423

The telehealth nurse found it easier to manage patients and carers whom she already knew well. She developed relationships through hospital visits and a telephone helpline and through liaison with the community MDT. When the telehealth nurse had less knowledge of the patient, it could be more difficult to interpret the information from TiM and liaison with the MDT was less easy. She also explained that she often did not know carers

well and sometimes felt there were less obvious solutions to carer distress making her reluctant to telephone. In contrast, most carers felt extremely positive that the nurse provided vital counselling and advice.

The telehealth nurse felt that a more formalised protocol, outlining how she should respond to alerts, would likely increase the number of telephone calls made to patients. However, she felt it was important that she could use her knowledge and experience to decide how to respond to the information presented and would find it difficult working to an overly prescriptive protocol.

### Improvements made to TiM during trial
Additional questions were added to the patient app: these recorded bladder and bowel symptoms and recorded disease progression in those with more severe disability (using the ALSFRS-EX[38]). Participants felt it was very important that the information provided was accurate and sometimes wanted to give more information in their answers. A free text facility was added, but this was used very infrequently. We added a heat map (figure 4) to the clinical portal, which displayed a visual representation of patients' answers over the previous 6 months. We added the ability for the telehealth nurse to temporarily pause alerts. The telehealth nurse found the heat map helpful to enable her to get an overview of things she thought were important, such as weight, and to see which category had changed. She also found the ability to pause alerts helpful.

### Recommended improvements to TiM
(Online supplementary data file tables 20–22)

The most important improvement we identified was to facilitate communication between the nurse and the participants. Most participants did not want automatic feedback on their progress from TiM: they could

judge their progress themselves. They were also worried how they would react to seeing objective measures of decline without any interpretation from their clinical team. However, many expected some feedback. The minimum expected was to know that their information had been received and reviewed and for their problems to be acknowledged in some way, even if no action was taken. Participants were asked about their preferences for communication. Telephone and face-to-face conversations were felt to be important for sensitive or complex discussions.

'I think it's not personal... I'd rather see somebody or talk to somebody than, than read about it on a screen.' Carer 366

Patients found email helped them communicate at their own pace. This was particularly helpful for those with speech problems where a phone call would be difficult. Participants were happy waiting for a definitive answer if they received an acknowledgement. Participants also thought TiM could be used to arrange appointments and other administrative matters and they were happy for their answers to be shared with other clinicians. The community MND nurse felt TiM information would be useful but the clinical portal would be difficult to access in the community without a computer. She also explained she was 'not good with computers', but would be happy receiving alerts and information from someone else.

## DISCUSSION

This detailed study of a digitally enabled MND service confirmed TiM was acceptable and feasible for patients, carers and staff. Differences in outcomes between the two groups were not formally assessed; however, our findings suggest that the TiM was accurate and safe. The study also gained a deeper understanding of the potential value of TiM and the context in which benefits might be seen.[39] We identified some key barriers and enablers to acceptance and use of telehealth, which are applicable to other digitally enabled care systems. A key initial barrier for participants was their perceived ability to master the technology. However, uptake was good with most participants (even those with severe disabilities) finding that they were able to use the system and sustain a good level of TiM adherence. The interviews suggested that this was because of the accessible software and face-to-face training provided. This is in contrast to other telehealth projects, which found acceptance and adherence were major barriers.[40] This study aligns with other evidence that suggests that patients have a positive attitude to digital products and, in the right setting, respond positively to digitally enabled care, as long as the service is accessible and continues to offer personal contact with healthcare professionals.[41 42] We recommend that new interventions should adopt our user-centred design process and offer additional support to those without confidence in technology to make the intervention accessible.

The potential impacts of digitally enabled care may differ depending on a patient's needs. TiM offered those with early or slowly progressive disease reassurance, information and the ability to remain in contact with the care team when they needed it as well as offering an alternative to clinic visits. Early identification and coordinated management of complications of the disease appear to be most valuable to those patients at a later stage of the disease or those with more rapidly changing disease. These factors may explain why MDC care improves survival. The study also highlighted the potential value of TiM to carers who could receive support separately from the patient, while still allowing them to fulfil their role as a carer. Patients and carers already use digital technology to seek reliable and relevant information in order to self-manage,[41] and the TiM can be used to provide information and promote self-management and self-efficacy (something that has a positive influence of patient quality of life[43]). This type of interaction also promotes good adherence to telehealth.[44] It was therefore interesting to find that participants in this study reported that they did not use the additional education services on TiM that frequently. This may be because many patients wish to deal with problems as they occur rather than learning extensively about their condition.[41] Telehealth could address this by providing a more personalised information resource, signposting to topics at pace appropriate for the user according to their circumstances and the answers they provide on TiM.

The main weakness of TiM was the mismatch between staff actions and the expectations of participants. This was due to lack of communication and high numbers of unnecessary alerts generated by TiM. MND progression is experienced as a series of losses,[45 46] and it is important for staff to acknowledge these problems and recognise the value that interactions with clinicians offer to patients and carers. Remote technology risks disrupting the traditional way in which the nurse would gather information, develop relationships and respond to patient difficulties.[47] There is a risk of care becoming less patient centre and patients becoming demoralised as the aims of patients/carers and the clinical team no longer align. We recommend care is taken to develop these interpersonal relationships and for staff to be aware of the potential pitfalls of remote management. Key improvements include improving the algorithms to reduce alerts, facilitating feedback and better communication between patients, carers and staff using technology (such as chat facilities, email or telemedicine) and better training of staff who are required to work in ways that differ from the traditional face-to-face model.

The key strength of this study was the depth in which we examined the processes occurring in order to explain not just what happened, but why events occurred and how TiM could be improved and implemented. However, this was a small study involving only one centre. As some patients died or withdrew early in the study, of the 20 assigned the TiM a maximum of 17 patients were using

the TiM at any one time. Only a small number of clinicians, many of whom had been helped with the TiM development, were involved. In particular, only one telehealth nurse used TiM, and this study showed that clinicians' behaviour clearly influences the way in which the intervention is delivered. In this case, a nurse who engaged more with the system may have resulted in better outcomes. Larger studies are needed to gain a true understanding of what would occur when the service is offered to all patients as 'business as usual', in different centres using different staff members. Other staff may behave differently, and our results are a reminder that the way they use the TiM warrants monitoring to ensure intervention fidelity and safety. The other limitations of evaluating complex interventions such as telehealth in traditional trials are discussed in detail in the parallel paper.[33] We recommend that future evaluations should retain the methods employed in this trial to understand processes in detail and examine implementation in different contexts, continuing to examine factors such as acceptability, adoption, fidelity, safety, costs and sustainability.

## CONCLUSION

The TiM system was acceptable to patients and carers with good uptake and adherence. Key to its success will be continued user and staff engagement and ensuring the system complements the existing services. Improvements to TiM should be based on the findings of this evaluation. Future process evaluations should also examine the implementation of TiM at other sites to determine whether the findings from this study can be generalised.

**Author affiliations**
[1]Sheffield Institute for Translational Neuroscience, University of Sheffield, Sheffield, UK
[2]Department of Neurology, Sheffield Teaching Hospitals NHS Trust, Sheffield, UK
[3]School of Health and Related Research, University of Sheffield, Sheffield, UK
[4]Clinical Trials Research Unit, School of Health and Related Research, University of Sheffield, Sheffield, UK
[5]Sheffield Motor Neurone Disease Association Research Advisory Group, Sheffield, UK

**Acknowledgements** This paper presents independent research funded by the National Institute for Health Research.

**Contributors** The following were involved: intervention development (CJMD, EH, TW, PJS), trial design and management (EH, MB, WB, CC, AQ, CJMD), clinical oversight (CJMD and PJS), recruitment and intervention delivery (EH and TW), data collection (EH, WB), data analysis (EH, MB, WB, CJMD, CC, SM, AQ) and manuscript preparation (all authors). All authors reviewed and commented on the manuscript. The authors are grateful to Sheffield MND Research Advisory Group for their guidance on the research methods.

**Funding** The TiM was developed within a collaboration between the University of Sheffield, Sheffield Teaching Hospitals NHS Trust, Mylan Ltd and Abbott Healthcare. This trial was funded by a National Institute for Health Research (NIHR) Doctoral Research Fellowship grant to EH (DRF-2013-06-076) and the Motor Neuron Disease Association. The trial was supported by the Sheffield Teaching Hospitals NHS Trust NIHR Clinical Research Facility and the University of Sheffield Clinical Trials Unit. Mylan Ltd supplied the software and hardware required for the trial. EH is also funded by an NIHR Clinical Lecturer award. PJS is supported by NIHR Senior Investigator award NF-SI-0512-10082 and PJS and CMD by the NIHR Sheffield Biomedical Research Centre (Translational Neuroscience) IS-BRC-1215-20017.

**Disclaimer** The views expressed are those of the author(s) and not necessarily those of the NHS, the NIHR or the Department of Health.

**Competing interests** The TiM intellectual property is owned by Mylan and the University of Sheffield.

**Patient consent for publication** Not required.

**Ethics approval** Approval was gained from Leeds Bradford Research Ethics Committee (REC reference 14/YH/1068) and the sponsor (Sheffield Teaching Hospitals NHS Foundation Trust Clinical Research Office).

**Provenance and peer review** Not commissioned; externally peer reviewed.

**Data availability statement** Data are available upon reasonable request.

**ORCID iDs**
Esther Hobson http://orcid.org/0000-0002-8497-2338
Wendy Baird http://orcid.org/0000-0002-4253-2721
Mike Bradburn http://orcid.org/0000-0002-3783-9761
Cindy Cooper http://orcid.org/0000-0002-2995-5447
Susan Mawson http://orcid.org/0000-0003-2795-89
Pamela J Shaw http://orcid.org/0000-0002-8925-2567
Christopher J McDermott http://orcid.org/0000-0002-1269-9053

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
