## [Reviewer comments · BMJ Open]

ARTICLE DETAILS

TITLE (PROVISIONAL)	A process evaluation and exploration of telehealth in motor neurone disease in a UK specialist centre
AUTHORS	Hobson, Esther; Baird, Wendy; Bradburn, Mike; Cooper, Cindy; Mawson, Susan; Quinn, Ann; Shaw, Pamela; Walsh, Theresa; McDermott, Christopher

VERSION 1 – REVIEW

REVIEWER	Sabrina Paganoni Massachusetts General Hospital and Spaulding Rehabilitation Hospital, Boston, MA- USA
REVIEW RETURNED	10-Jan-2019

GENERAL COMMENTS	This is a very well-written paper about an important and timely topic. The authors are leading the development of an innovative tool for ALS care and, in reading this and other papers by this group, it is clear that they are following a systematic and thoughtful iterative process. It is wonderful to see that they are incorporating feedback from multiple stakeholders and they are leveraging real-world experience from pilot studies to enhance their product. Insights gained from the present study will continue to inform the development of TIM and improve the quality of their work. I was surprised to see that the authors do not comment on the evolving ALS telehealth landscape (outside of their institution). Other applications are now being used for ALS care and research. While I understand this paper is not a review on the topic, it may still be helpful to at least cite the work of other groups that are now using telehealth for ALS care as "business as usual". This includes videoconference tools that are now used routinely for ALS care in the US (see the VA experience [PMID: 28427284] and other recent papers [PMID: 29250986;PMID: 28678542]). It may also be helpful to cite ongoing efforts to develop other apps for ALS care and research (ALS at Home, Answer ALS app, Beiwe app, Prize4Life app). As different telehealth tools are developed, it will be important to understand how they compare as they may have different use cases, cost-effectiveness profiles, and impact on health outcomes. The introductory sentence "no system can monitor the wide range of problems faced by patients with MND" (Introduction, line 31) is a blanket statement that does not take into the account the work that is ongoing at other institutions.
--

REVIEWER	Anne Hogden University of Tasmania, Australia
REVIEW RETURNED	22-Jan-2019

GENERAL COMMENTS	It was pleasing to read this manuscript on the evaluation of an intervention that offers a strong contribution to MND patient care. It's great to see more of this study coming into publication. I note that this manuscript is a parallel paper with a feasibility study, submitted as separate manuscripts. While there are several overlaps, I believe both manuscripts work well as stand-alone papers. I reviewed the feasibility manuscript first, and some of the comments made apply to both manuscripts. My comments aim to improve the readability of the manuscript. It is well structured and generally well presented, but also contains many editing errors throughout. I recommend a tighter edit of the manuscript overall. Background: TiM needs to be written in full for it's first use in the main text (paragraph 2) Aims: The aims statements are very complicated, and challenging to digest. It would be easier to read and remember the aims if the language was simplified, and each aim was explicitly linked to the subheadings in the results section, to show how each of the aims were met. For example, by numbering each aim, and reducing the excess wording, particularly in the first aim. Data collection: As there is so much supplementary data, it would help the reader if the main text contains a reference to the appropriate supplementary table. Public and Patient involvement: A big chunk of this section is identical to that of the feasibility paper. Please rephrase the repeated sentences to avoid the complications this creates. Results: The results are well-structured and cover many of the challenges that could be expected with e-health delivery. It was good to hear the patient and carer voice throughout this section. The capacity of the system to allow early identification if problems is an encouraging finding that has implications for proactive care and timely decision-making. Table 2 is a useful way to demonstrate practical solutions to a range of technology problems, and their impact on care. This will be helpful to those setting up e-health systems. Discussion: The key findings are expanded well. The first 2 sentences create a little confusion if the two papers are read in parallel. I suggest combining the two sentences by removing 'As this was a feasibility study' . Please consider adding a second reference to your supporting evidence for patient support for telehealth in MND, in addition to face to face care - James et al (2018) Patients' perspectives of multidisciplinary home-based e-Health service delivery for motor neurone disease, Disability and Rehabilitation: Assistive Technology, DOI: 10.1080/17483107.2018.1499139
---

	Paragraph 3 discusses the main barrier to successful TiM implementation. Is there capacity to improve how staff are trained to use TiM, and to understand that what is important to patients and carers should also be important to them? The beauty of remote monitoring is that it allows clinicians to acknowledge patient's concerns in a timely way, even if no action is taken, and helps to manage patient's anxiety in real time. This paragraph needs some careful editing. The third sentence needs rephrasing to improve clarity - '.. and recognise the value they offer to patients...' - does 'they' refer to the clinicians?
--	---

REVIEWER	Michael Pulley University of Florida, Jacksonville. USA
REVIEW RETURNED	31-Jan-2019

GENERAL COMMENTS	The authors planned to evaluate their method of telecare (TiM) in a randomized, controlled study. This paper only addresses the intervention arm of the study but does not compare the intervention to usual care and contrast differences regarding patient attitudes toward care. I assume this is addressed in the accompanying feasibility study. The paper does not acknowledge the limitation of a single telehealth nurse being involved in the study and how, as the primary point of contact for all patient in the study, this individual had such a critical role in the patient's experience. I felt that based on patient replies and concern about lack of feedback, there should have been a more concerted effort to indicate at least acknowledgement that a trigger was noted and a brief message sent to the patient with the possibility that the patient could contact for further follow-up if desired. The authors used the term "bespoke" and this is not a term I was familiar with and perhaps should be changed to a more common term like personalized or customized. This use was in relation to the idea of providing educational materials. It might be more appropriate to have suggestions to the patient or caregiver to read about a topic when the alerts are triggered to provide information at a time that is appropriate for their stage of disease and avoid information overload or worry earlier in the course. The number of patients participating and who actually received TiM is confusing. It was stated that 40 patients and 37 caregivers participated but then it later said that only 17 were participating at any given time. Was 40 the total and were 17 assigned to TiM and 23 assigned to usual care? Please clarify the numbers for better understanding. More references to the findings from the feasibility study in this paper would be helpful for the reader to clarify some things about the exact process, questions asked, algorithm for the telehealth nurse, etc
---

REVIEWER	Palmira Bernocchi, PhD ICS Maugeri, Italy
REVIEW RETURNED	27-Feb-2019

GENERAL COMMENTS	This work is linked to the previous one. It presents a very accurate and qualitative analysis of the data to demonstrate the usability of the model TiM. The authors state in the brackground "that" few interventions offer support to informal career"referring to no works. In reality there are
---

	some works that should be reported such as : Amyotroph Lateral Scler Frontotemporal Degener. 2015 Jun;16(3-4):187-95. doi: 10.3109/21678421.2014.974616. Epub 2014 Nov 27. or BMJ Open. 2018 Jan 27;8(1):e018721. doi: 10.1136/bmjopen-2017-018721. or J Telemed Telecare. 2010;16(2):83-8. doi: 10.1258/jtt.2009.090604. Epub 2010 Feb 5. Aim Also in this work as in the previous one we speak of a randomized and controlled study. But no information is reported regarding what the controls do. Some information would be useful. For a better reading of the work, both the methods and the results should be structured in order to better identify what has been done and what are the specific results that identify the three aims reported by the authors:  1. Explore TiM implementation, fidelity and quality, and acceptability and Feasibility of using to deliver specialist, multidisciplinary care at a distance 2. Clarify potential mechanisms of impact (both intended and unitended) 3 Identify contextual factors that might influence the implementation or impact 4.... Using the same descriptive modality of aims in both methods and results helps to read the work Results In table 1 I did not understand what use do the controls of the TiM app. What is the difference between the two groups in the use of telehealth Discussion Describe better the limitations of this study. It is commendable the great work done by the authors in patients with a rare and debilitating pathology that require a considerable use of health and social services. It is important, however, to make the reading of the two works "easier". It is important that the two papers be as independent as possible one from the other, avoiding if possible to continue recalling parts described in the other work.
--	---

VERSION 1 – AUTHOR RESPONSE

Reviewer: 1

Reviewer Name: Sabrina Paganoni

Institution and Country: Massachusetts General Hospital and Spaulding Rehabilitation Hospital, Boston, MA- USA

Please state any competing interests or state 'None declared': None declared

Please leave your comments for the authors below

This is a very well-written paper about an important and timely topic. The authors are leading the development of an innovative tool for ALS care and, in reading this and other papers by this group, it is clear that they are following a systematic and thoughtful iterative process. It is wonderful to see that they are incorporating feedback from multiple stakeholders and they are leveraging real-world experience from pilot studies to enhance their product. Insights gained from the present study will continue to inform the development of TiM and improve the quality of their work.

We thank the reviewer for their positive comments.

I was surprised to see that the authors do not comment on the evolving ALS telehealth landscape (outside of their institution). Other applications are now being used for ALS care and research. While I understand this paper is not a review on the topic, it may still be helpful to at least cite the work of other groups that are now using telehealth for ALS care as "business as usual". This includes videoconference tools that are now used routinely for ALS care in the US (see the VA experience [PMID: 28427284] and other recent papers [PMID: 29250986; PMID: 28678542]). It may also be helpful to cite ongoing efforts to develop other apps for ALS care and research (ALS at Home, Answer ALS app, Beiwe app, Prize4Life app). As different telehealth tools are developed, it will be important to understand how they compare as they may have different use cases, cost-effectiveness profiles, and impact on health outcomes. The introductory sentence "no system can monitor the wide range of problems faced by patients with MND" (Introduction, line 31) is a blanket statement that does not take into the account the work that is ongoing at other institutions.

We have added in references to both telemedicine and telehealth as suggested and also reference our own systemic review of the area.

Reviewer: 2

Reviewer Name: Anne Hogden

Institution and Country: University of Tasmania, Australia

Please state any competing interests or state 'None declared': None declared

Please leave your comments for the authors below

It was pleasing to read this manuscript on the evaluation of an intervention that offers a strong contribution to MND patient care. It's great to see more of this study coming into publication. I note that this manuscript is a parallel paper with a feasibility study, submitted as separate manuscripts. While there are several overlaps, I believe both manuscripts work well as stand-alone papers. I reviewed the feasibility manuscript first, and some of the comments made apply to both manuscripts.

My comments aim to improve the readability of the manuscript. It is well structured and generally well presented, but also contains many editing errors throughout. I recommend a tighter edit of the manuscript overall.

This has been done.

Background:

TiM needs to be written in full for it's first use in the main text (paragraph 2)

This has been added.

Aims:

The aims statements are very complicated, and challenging to digest. It would be easier to read and remember the aims if the language was simplified, and each aim was explicitly linked to the subheadings in the results section, to show how each of the aims were met. For example, by numbering each aim, and reducing the excess wording, particularly in the first aim.

The wording has been amended as suggested and now reflects the subheadings in the results

“We aimed to observe the processes that occurred when the TiM was used to deliver specialist multidisciplinary care in MND exploring:

- TiM technology set-up and delivery
- Participants’ adherence to TiM
- Participants’ attitudes towards TiM
- Clinicians’ attitudes towards TiM
- Potential impacts of the TiM on participants and staff (both intended and unintended)
- The mechanisms and contextual factors that may affect the impact and implementation on a larger scale.

We also aimed to:

- Identify problems with TiM
- Begin to implement and evaluate improvements
- Identify further improvements and uses of the TiM.

Data collection:

As there is so much supplementary data, it would help the reader if the main text contains a reference to the appropriate supplementary table.

These have been added

Public and Patient involvement:

A big chunk of this section is identical to that of the feasibility paper. Please rephrase the repeated sentences to avoid the complications this creates.

This was requested by the editorial team at the time of submission. We will be guided by them as to whether to have substantial overlap as requested or for one paper to refer to the other.

Results:

The results are well-structured and cover many of the challenges that could be expected with e-health delivery. It was good to hear the patient and carer voice throughout this section. The capacity of the system to allow early identification if problems is an encouraging finding that has implications for proactive care and timely decision-making.

Table 2 is a useful way to demonstrate practical solutions to a range of technology problems, and their impact on care. This will be helpful to those setting up e-health systems.

Discussion:

The key findings are expanded well.

The first 2 sentences create a little confusion if the two papers are read in parallel. I suggest combining the two sentences by removing 'As this was a feasibility study' .

We have removed this.

Please consider adding a second reference to your supporting evidence for patient support for telehealth in MND, in addition to face to face care - James et al (2018) Patients’ perspectives of

multidisciplinary home-based e-Health service delivery for motor neurone disease, Disability and Rehabilitation: Assistive Technology, DOI: 10.1080/17483107.2018.1499139

We have added this reference.

Paragraph 3 discusses the main barrier to successful TiM implementation. Is there capacity to improve how staff are trained to use TiM, and to understand that what is important to patients and carers should also be important to them? The beauty of remote monitoring is that it allows clinicians to acknowledge patient's concerns in a timely way, even if no action is taken, and helps to manage patient's anxiety in real time.

We added:

“Key improvements include improving the algorithms to reduce alerts, facilitating better communication between patients, carers and staff using both technology (such as chat facilities, email or telemedicine). Better training of staff is also needed when working in services that differ from the traditional face-to-face model.”

This paragraph needs some careful editing. The third sentence needs rephrasing to improve clarity - '.. and recognise the value they offer to patients...' - does 'they' refer to the clinicians?

Amended to “and it is important for staff to acknowledge these problems and recognise the value that interactions with clinicians offer to patients and carers.”

Reviewer: 3

Reviewer Name: Michael Pulley

Institution and Country: University of Florida, Jacksonville. USA

Please state any competing interests or state 'None declared': None declared

Please leave your comments for the authors below

The authors planned to evaluate their method of telecare (TiM) in a randomized, controlled study. This paper only addresses the intervention arm of the study but does not compare the intervention to usual care and contrast differences regarding patient attitudes toward care. I assume this is addressed in the accompanying feasibility study. The paper does not acknowledge the limitation of a single telehealth nurse being involved in the study and how, as the primary point of contact for all patient in the study, this individual had such a critical role in the patient's experience. I felt that based on patient replies and concern about lack of feedback, there should have been a more concerted effort to indicate at least acknowledgement that a trigger was noted and a brief message sent to the patient with the possibility that the patient could contact for further follow-up if desired.

The reviewer's point about the telehealth nurse has been noted and added this into both the discussion and strengths and limitations section.

“However, this was a small study involving only one centre. As some patients died or withdrew early in the study, of the 20 assigned the TiM a maximum of 17 patients were using the TiM at any one time. Only a small number of clinicians who had been involved with the TiM development were involved. In particular, only one telehealth nurse used TiM and their behaviour clearly influenced the way in which the intervention was delivered. Larger studies are needed to gain a true understanding of what would occur when the service is offered to all patients as “business as usual”, in different centres using different staff members. Other staff may behave differently and our results are a

reminder that they way they use the TiM warrants monitoring to ensure intervention fidelity and safety.”

The problems about automatic feedback are addressed in Recommend improvements to TiM section and in the discussion

“The most important improvement we identified was to facilitate communication between the nurse and participants. Most participants did not want automatic feedback on their progress from TiM: they could judge their progress themselves. They were also worried how they would react to seeing objective measures of decline without any interpretation from their clinical team. However, many expected some feedback. The minimum expected was to know that their information had been received and reviewed and for their problems to be acknowledged in some way, even if no action was taken.”

“Key improvements include improving the algorithms to reduce alerts, facilitating feedback and better communication between patients, carers and staff using both technology (such as chat facilities, email or telemedicine).”

The authors used the term "bespoke" and this is not a term I was familiar with and perhaps should be changed to a more common term like personalized or customized.

Amended to “personalised”

This use was in relation to the idea of providing educational materials. It might be more appropriate to have suggestions to the patient or caregiver to read about a topic when the alerts are triggered to provide information at a time that is appropriate for their stage of disease and avoid information overload or worry earlier in the course.

Amended as suggested.

“Telehealth could address this by providing a more personalised information resource with topics and pace appropriate for the user according to their circumstances and the answers they provide on TiM.”

The number of patients participating and who actually received TiM is confusing. It was stated that 40 patients and 37 caregivers participated but then it later said that only 17 were participating at any given time. Was 40 the total and were 17 assigned to TiM and 23 assigned to usual care? Please clarify the numbers for better understanding.

We have clarified this sentence:

“As some patients died or withdrew early in the study, of the 20 assigned the TiM a maximum of 17 patients were using the TiM at any one time.”

More references to the findings from the feasibility study in this paper would be helpful for the reader to clarify some things about the exact process, questions asked, algorithm for the telehealth nurse, etc

If both papers are accepted we will request references to each paper are inserted throughout. We have also referenced the original TiM paper which described the TiM in detail. We have also included more references to the online data supplements which detail the methods and results (online supplements in square brackets, highlighted in yellow).

Reviewer: 4

Reviewer Name: Palmira Bernocchi, PhD

Institution and Country: ICS Maugeri, Italy

Please state any competing interests or state 'None declared': none declared

Please leave your comments for the authors below

This work is linked to the previous one. It presents a very accurate and qualitative analysis of the data to demonstrate the usability of the model TiM.

The authors state in the background "that" few interventions offer support to informal career"referring to no works. In reality there are some works that should be reported such as :

Amyotroph Lateral Scler Frontotemporal Degener. 2015 Jun;16(3-4):187-95. doi:

10.3109/21678421.2014.974616. Epub 2014 Nov 27.

or

BMJ Open. 2018 Jan 27;8(1):e018721. doi: 10.1136/bmjopen-2017-018721.

or

J Telemed Telecare. 2010;16(2):83-8. doi: 10.1258/jtt.2009.090604. Epub 2010 Feb 5.

Aim

We have added a paragraph discussing carers into the introduction.

Also in this work as in the previous one we speak of a randomized and controlled study. But no information is reported regarding what the controls do. Some information would be useful.

We have added this statement to the study design:

"In brief, patients with MND and their primary informal carer currently receiving care from the Sheffield MDC were recruited and randomised to receive either the intervention (telehealth plus usual care) or control (usual care alone). Usual care involved invitations to the multidisciplinary clinic two to six monthly plus access between visits to the multidisciplinary team via a MND specialist nurse, by telephone, email or through liaison with other healthcare progressions."

For a better reading of the work, both the methods and the results should be structured in order to better identify what has been done and what are the specific results that identify the three aims reported by the authors:

1. Explore TiM implementation, fidelity and quality, and acceptability and Feasibility of using to deliver specialist, multidisciplinary care at a distance
2. Clarify potential mechanisms of impact (both intended and unintended)
- 3 Identify contextual factors that might influence the implementation or impact
- 4....

Using the same descriptive modality of aims in both methods and results helps to read the work

This has been amended:

"We aimed to observe the processes that occurred when the TiM was used to deliver specialist multidisciplinary care in MND exploring:

- TiM technology set-up and delivery
- Participants' adherence to TiM
- Participants' attitudes towards TiM
- Clinicians' attitudes towards TiM
- Potential impacts of the TiM on participants and staff (both intended and unintended)

- The mechanisms and contextual factors that may affect the impact and implementation on a larger scale.

We also aimed to:

- Identify problems with TiM
- Begin to implement and evaluate improvements
- Identify future improvements and uses”.

Results

In table 1 I did not understand what use do the controls of the TiM app. What is the difference between the two groups in the use of telehealth

We have clarified in the study design that there was a control and intervention group, and described usual care

“In brief, patients with MND and their primary informal carer currently receiving care from the Sheffield MDC were recruited and randomised to receive either the intervention (TiM plus usual care) or control (usual care alone). Usual care involved invitations to the MDC two to six monthly plus access between visits to the multidisciplinary team via a MND specialist nurse by telephone, email or through liaison with other healthcare professionals.”

Discussion

Describe better the limitations of this study.

We have included additional limitations in the final paragraph of the discussion.

“However, this was a small study involving only one centre. As some patients died or withdrew early in the study, of the 20 assigned the TiM a maximum of 17 patients were using the TiM at any one time. Only a small number of clinicians who had been involved with the TiM development were involved. In particular, only one telehealth nurse used TiM and their behaviour clearly influenced the way in which the intervention was delivered. Larger studies are needed to gain a true understanding of what would occur when the service is offered to all patients as “business as usual”, in different centres using different staff members. Other staff may behave differently and our results are a reminder that they way they use the TiM warrants monitoring to ensure intervention fidelity and safety.”

It is commendable the great work done by the authors in patients with a rare and debilitating pathology that require a considerable use of health and social services.

It is important, however, to make the reading of the two works "easier".

It is important that the two papers be as independent as possible one from the other, avoiding if possible to continue recalling parts described in the other work.

We appreciate presenting the large amount of work is challenging and trying to provide sufficient information without length papers does pose difficulties. We have reviewed both papers together and tried to clarify some of the methods. We have also linked to the supplementary results data more clearly underneath each subheading and included the protocol and statistics analysis plan as an additional supplementary file (file references in square brackets)

VERSION 2 – REVIEW

REVIEWER	Sabrina Paganoni Massachusetts General Hospital and Spaulding Rehabilitation Hospital, Harvard Medical School, Boston, MA - USA
REVIEW RETURNED	24-May-2019

GENERAL COMMENTS	The authors addressed the reviewers' concerns and provided excellent additional material. A few additional suggestions: 1- Please avoid primacy statements such as: "This is the first evaluation of a digitally-enabled care system.... ". Several systems are in development and are being piloted at different institutions. While these efforts have not resulted in publications yet, it would be important to note that they are available including systems connected to electronic medical records that can send clinicians reminders/alerts in between visits. 2- Consider referencing more recent work in ALS telehealth such as PMID: 29802746
--

REVIEWER	Anne Hogden University of Tasmania, Australia
REVIEW RETURNED	03-Jun-2019

GENERAL COMMENTS	The authors have made a solid effort to meet the requirements of four reviewers. The manuscript now reads well, with clear aims and limitations stated. I look forward to seeing this manuscript published.
---

REVIEWER	Michael Pulley University of Florida, Jacksonville
REVIEW RETURNED	19-Jun-2019

GENERAL COMMENTS	Just some minor correction. In the Data Collection section: "To gain a deeper understanding of the processes involved we evaluated patient, carer and staff experiences of the TiM. We used satisfaction questionnaires, collected adverse"..... should read "collected adverse events"... I suppose In supplementary material there is a mistake : in the following sentence "participant" should read "participate": 6b. Consent Following indication of their interest to participant potential participants will be met at a mutually agreeable location, preferably the patients' home. The telehealth nurse seemed to become less engaged or perhaps too busy as the study went on and may have missed a patient alert for more than a week. "She explained that initially she looked at the system every day but by the end of the trial she looked weekly, and sometimes a little less often." "After seeing patients in their clinic using TiM, clinicians completed a feedback from" should read "form", not "from" "23 forms were completed." This seems like a shockingly low number given that: 40 patients and 37 informal carers (three patients
---

	had no carer) were recruited. Two TiM patients withdrew due to severe illness and one died before six months, the rest used TiM for between six and 18 months. Can the authors comment about the low number of forms filled out In the discussion: "The study also gained a deeper understanding of the potential value of TiM and the context in which benefits may might be seen " Use either may or might, not both "participants in this study did not use the additional education services on TiM that frequently. " The authors stated earlier that this was not actually tracked. Also, given the lack of use how do the authors then justify the statements "This may be because many patients wish to deal with problems as they occur, rather than learning extensively about their condition (36). Telehealth could address this by providing a more personalised information resource with topics and pace appropriate for the user according to their circumstances and the answers they provide on TiM." It seems that if the patients aren't accessing the information very often then they are not likely to address their specific situation. "There is a risk of care becoming less patient centre an patients becoming demoralised as the aims of patients/carers and the clinical team no longer align." should read "centered and" not "centre an"
--	--

VERSION 2 – AUTHOR RESPONSE

Reviewer: 1

Reviewer Name: Sabrina Paganoni

Institution and Country: Massachusetts General Hospital and Spaulding Rehabilitation Hospital, Harvard Medical School, Boston, MA - USA

Please state any competing interests or state 'None declared': None Declared

Please leave your comments for the authors below

The authors addressed the reviewers' concerns and provided excellent additional material. A few additional suggestions:

1- Please avoid primacy statements such as: "This is the first evaluation of a digitally-enabled care system.... ". Several systems are in development and are being piloted at different institutions. While these efforts have not resulted in publications yet, it would be important to note that they are available including systems connected to electronic medical records that can send clinicians reminders/alerts in between visits.

Changed to:

"This is a detailed evaluation of a digitally-enabled care system that aims to provide accessible, specialist holistic care to patients with motor neuron disease and their carers."

2- Consider referencing more recent work in ALS telehealth such as PMID: 29802746

We have added this reference.

Reviewer: 2

Reviewer Name: Anne Hogden

Institution and Country: University of Tasmania, Australia

Please state any competing interests or state 'None declared': None declared

Please leave your comments for the authors below

The authors have made a solid effort to meet the requirements of four reviewers. The manuscript now reads well, with clear aims and limitations stated.

I look forward to seeing this manuscript published.

Reviewer: 3

Reviewer Name: Michael Pulley

Institution and Country: University of Florida, Jacksonville

Please state any competing interests or state 'None declared': None declared

Please leave your comments for the authors below

Just some minor correction.

In the Data Collection section:

"To gain a deeper understanding of the processes involved we evaluated patient, carer and staff experiences of the TiM. We used satisfaction questionnaires, collected adverse"..... should read "collected adverse events"... I suppose

This has been amended as suggested.

In supplementary material there is a mistake : in the following sentence "participant" should read "participate":

6b. Consent

Following indication of their interest to participant potential participants will be met at a mutually agreeable location, preferably the patients' home.

We believe this refers to the study protocol. It would not be possible to change this at this stage.

The telehealth nurse seemed to become less engaged or perhaps too busy as the study went on and may have missed a patient alert for more than a week. "She explained that initially she looked at the system every day but by the end of the trial she looked weekly, and sometimes a little less often."

It was not possible to determine whether the nurse missed alerts, we do describe in detail that there were too many alerts and this will have had an impact on her behaviour (see section TiM system alerts). We have added this statement to the discussion:

"Only a small number of clinicians, many of whom had been helped with the TiM development were involved. In particular, only one telehealth nurse used TiM and this study showed that clinicians' behaviour clearly influences the way in which the intervention is delivered.

STATEMENT ADDED: In this case, a nurse who engaged more with the system may have resulted in better outcomes.

Larger studies are needed to gain a true understanding of what would occur when the service is offered to all patients as “business as usual”, in different centres using different staff members. Other staff may behave differently and our results are a reminder that they way they use the TiM warrants monitoring to ensure intervention fidelity and safety. “

After seeing patients in their clinic using TiM, clinicians completed a feedback form" should read "form", not "from"

This has been corrected.

"23 forms were completed." This seems like a shockingly low number given that: 40 patients and 37 informal carers (three patients had no carer) were recruited. Two TiM patients withdrew due to severe illness and one died before six months, the rest used TiM for between six and 18 months. Can the authors comment about the low number of forms filled out

Forms were only completed for patients using the TiM system. A total of 20 patients used the TiM system. Some never attended clinic during the trial because they were too unwell. We have mentioned in the limitations section that feedback was gained from only a small number of clinicians and this warrants further evaluation in future studies.

“Only a small number of clinicians, many of whom had been helped with the TiM development were involved. In particular, only one telehealth nurse used TiM and this study showed that clinicians’ behaviour clearly influences the way in which the intervention is delivered. In this case, a nurse who engaged more with the system may have resulted in better outcomes. Larger studies are needed to gain a true understanding of what would occur when the service is offered to all patients as “business as usual”, in different centres using different staff members. Other staff may behave differently and our results are a reminder that they way they use the TiM warrants monitoring to ensure intervention fidelity and safety.”

In the discussion: "The study also gained a deeper understanding of the potential value of TiM and the context in which benefits may might be seen " Use either may or might, not both

This mistake has been corrected to

"The study also gained a deeper understanding of the potential value of TiM and the context in which benefits might be seen”

"participants in this study did not use the additional education services on TiM that frequently. " The authors stated earlier that this was not actually tracked. Also, given the lack of use how do the authors then justify the statements "This may be because many patients wish to deal with problems as they occur, rather than learning extensively about their condition (36). Telehealth could address this by providing a more personalised information resource with topics and pace appropriate for the user according to their circumstances and the answers they provide on TiM." It seems that if the patients aren't accessing the information very often then they are not likely to address their specific situation.

We have amended this sentence to clarify:

It was therefore interesting to find that participants in this study reported that they did not use the additional education services on TiM that frequently.

We have amended the next statement to mention signposting. We believe that the telehealth system could signpost patients to access information. They would complete the weekly telehealth system and

be directed by the telehealth app to access information specific to their needs.

Telehealth could address this by providing a more personalised information resource, signposting to topics at pace appropriate for the user according to their circumstances and the answers they provide on TiM.

"There is a risk of care becoming less patient centre an patients becoming demoralised as the aims of patients/carers and the clinical team no longer align." should read "centered and" not "centre an"

Corrected